# What influences engagement with a bipolar disorder self-management app? A qualitative investigation of use of the PolarUs app

Emma Morton[1], Rachelle Hole[2], Heather O'Brien[3], Linda Li[4], Steven J. Barnes[5], Erin E. Michalak[6]*

1 School of Psychological Sciences, Monash University, Clayton, Victoria, Australia, 2 School of Social Work, University of British Columbia, Okanagan, British Columbia, Canada, 3 School of Information, University of British Columbia, Vancouver, British Columbia, Canada, 4 Department of Physical Therapy, University of British Columbia, Vancouver, British Columbia, Canada, 5 Department of Psychology, University of British Columbia, Vancouver, British Columbia, Canada, 6 Department of Psychiatry, University of British Columbia, Vancouver, British Columbia, Canada

* erin.michalak@ubc.ca

## Abstract

Interventions delivered via smartphone apps may support individuals with bipolar disorder (BD) to learn about and implement evidence-based self-management strategies in the context of their daily lives. However, app usage rates are often suboptimal. The subjective experience of users may provide insights into factors influencing engagement (and disengagement) with an mHealth intervention. The present study describes a qualitative investigation of the experiences of people with BD who participated in the evaluation of a novel app-based intervention for BD self-management, the PolarUs app. Twenty-five individuals with BD were provided with access to an app-based self-management intervention over a three-month study period, and were later interviewed about personal experiences of engagement with the intervention, including attempts to enact self-management strategies. Thematic analysis was used to identify important aspects of the experience of engaging with a self-management app. Three themes describing drivers of engagement with the PolarUs app and associated features were generated: 1) Motivations, 2) Salience, and 3) Perceived effort. Drivers of engagement were shaped by contextual influences, summarised in four themes: 1) The smartphone ecosystem, 2) Daily life, 3) Mood symptoms, and 4) Involvement in a research study. The findings of this research generate insights into how individuals with BD engage with app-based interventions. Lived experience perspectives can inform the design of engaging app-based interventions for BD. Further, these findings emphasise the importance of considering the context in which people use self-management apps for BD for both research studies and implementation.

which permits unrestricted use, distribution, and reproduction in any medium, provided the original author and source are credited.

**Data availability statement:** As this is a qualitative study, the data set involves richly detailed interviews where individuals spoke about their life circumstances and experiences. Even if potentially identifying details like names, ages, and locations were removed from the data set, it is still possible that these interviews, when viewed in full, contain enough information to identify the participants. The ethical protocol for this study restricts public sharing of some data sets; as such, these are available by request from the University of British Columbia Clinical Research Ethics Boards. Interested investigators may submit inquiries to the corresponding author. The study was approved by the University of British Columbia Behavioural Research Ethics Board. Contact details are as follows: UBC BREB Office #102, Technology Enterprise Facility (TEF) III 6190 Agronomy Road Vancouver, BC V6T 1Z3 Fax: 604-822-5093 allison.stewartgomes@ubc.ca.

**Funding:** This research was supported by a Canadian Institutes of Health Research (https://cihr-irsc.gc.ca/e/193.html) Project Grant (Grant number: 419504; Principal Applicants: EEM, SJB, Co-Applicants: EM, RH, HOB). EM received salary support from a Canadian Institutes of Health Research Banting Postdoctoral Fellowship (Award number: 453869). The funder had no role in study design, data collection and analysis, decision to publish, or preparation of the manuscript.

**Competing interests:** I have read the journal's policy and the authors of this manuscript have the following potential competing interests: EM has received honorarium for advising on the development of educational materials for Neurotorium, an online educational platform supported by the Lundbeck Foundation. EEM has received funding to support unrelated patient education initiatives from Otsuka-Lundbeck. RH, HOB, LL, and SJB declared no potential competing interests with respect to the research, authorship, and/or publication of this article.

## Author summary

People with bipolar disorder may benefit from using mental health apps to keep track of their symptoms and learn about ways to manage their health. However, many people find it difficult to keep using apps for long periods of time. In this study, we interviewed 25 people who were given the opportunity to use a new app for bipolar disorder (PolarUs) over a three-month period. We asked them what influenced their use of this app. People told us that a number of factors influenced whether or not they used the app: their motivation levels, their awareness of the app, and the level of effort involved. They also said that the context in which they used the app mattered: other apps on their smartphone could be a distraction, or they were too busy with other tasks to use the app. Sometimes bipolar disorder symptoms made it more difficult to use the app. Some people kept using the app because they wanted to help the research. These findings can help other developers create apps that people will be more likely to use over a long period of time.

## Introduction

Digital health interventions have potential to support individuals with bipolar disorder (BD) to access and implement evidence-based self-management strategies to improve their quality of life (QoL), reduce mood symptoms, and maintain mood stability [1,2]. A survey in the UK indicated that more than 90% of people with BD own a smartphone [3], and surveys in the UK, US, and internationally reported between 42–77% are interested in (or indeed, already using) app-based self-management features [3–5]. Indeed, recent years have seen a rapid proliferation of mental health apps, both from commercial and research developers, that claim to assist with various BD-related self-management tasks [6,7].

A noted challenge limiting the potential of mental health apps in general is their ability to sustain user engagement, typically defined according to usage behaviours such as frequency and duration of use and measured through logins and clicks/views for app features and content [8]. Although not specific to BD, an analysis of app store metrics found that 15 days after download, user retention in mental health apps was only 4%, and by 30 days that had further dropped to 3.3% [9]. Engagement with apps in the context of research trials is similarly sub-optimal; for example, drop-out rates in randomised control trials of smartphone interventions for depression were almost 50% after accounting for publication bias [10]. Less research is available to quantify engagement rates in app-based interventions for BD; a review of evaluations to date suggested participants used apps on 58–91% of possible occasions [11]. In addition, adherence characteristics were sometimes reported relative to active participants only, masking drop-out rates. Given such heterogenous reporting of adherence, we cannot conclusively infer whether or not BD app interventions have successfully addressed engagement-related challenges. Greater understanding of if, and how,

PLOS Digital Health

individuals with BD use mental health apps is needed to develop effective interventions and plan for their successful implementation in real-world contexts.

Evaluations of app engagement based on usage metrics alone may not be sufficient to capture the complex behavioural, cognitive, and emotional processes at play [12]. For example, it is typically presumed that higher usage rates will be associated with greater therapeutic impacts, yet an evaluation of the MONARCA self-monitoring BD app found that despite adherence rates of over 93%, individuals in the intervention arm had worse depressive symptoms [13]. Usage data alone cannot illuminate whether increased app use is driven by worsening mental health, or vice versa. Usage metrics may also not be able to identify or explain non-clinically beneficial engagement – for example, gamification (the use of game elements such as challenges, streaks, or in-app rewards) has been commonly suggested as a feature capable of supporting user retention [14], yet poorly considered gamification features may siphon engagement away from more clinically relevant aspects [15]. To understand why individuals engage or disengage with an app, and how engagement relates to therapeutic outcomes, investigation of the subjective experience of app use is required.

Some evaluations of digital interventions have used quantitative survey methods to explore reasons for disengagement, e.g., the self-management focused Life Goals app [16]. However, survey-based research may not be able to adequately illuminate the complex and intersecting influences on app engagement and disengagement. Qualitative research is therefore needed for in-depth exploration of these subjective processes [12], however few evaluations of mental health apps for BD have included a qualitative component [17]. The qualitative evaluation of the LiveWell app focused on user experiences of the behaviour change process and not engagement specifically [18], while the qualitative evaluation of the Life Goals app focused on perceptions of app content and features [19].

To advance understanding in this area, the present study therefore used qualitative interviews to explore prominent aspects of the experience of engaging with a novel app-based self-management intervention for BD, the PolarUs app [20]. Our research aimed to both understand self-reported patterns of engagement with the app over time, as well as factors (including both app features and other salient influences) which were perceived by users to influence engagement.

## Results

### Participants

Of participants in the PolarUs study who indicated consent to be contacted, 30.8% ($n = 48$) were invited to participate; twenty-five individuals (52.1% of invited participants) provided consent and were interviewed (see Methods for a detailed description of sampling). Interviews ranged from 25.6 to 61.2 minutes ($M = 46.2$ minutes, $SD = 9.4$), and were conducted between 9 and 52 days following the conclusion of the intervention ($M = 28.3$ days, $SD = 12.3$). Interviews were conducted between November, 2021 and March, 2022. Sample characteristics are described in Table 1; 60% ($n = 15$) were women, 92% ($n = 23$) had a diagnosis of BD-I, 64% ($n = 16$) reported White/European ethnicity, 56% ($n = 14$) resided in Canada, 48% ($n = 12$) were engaged in full time work, and the majority (88%; $n = 22$) had completed some form of post-secondary education. The mean age of the sample was 38.3 years ($SD = 11.5$).

### Overview of key findings

Factors that influenced engagement with the PolarUs app were grouped under an overarching theme: Drivers of engagement, which included three sub-themes: 1) Motivations, 2) Salience, and 3) Perceived effort. Factors that interacted with drivers of engagement were grouped under an overarching theme: Contextual influences, which included four sub-themes: 1) The smartphone ecosystem, 2) Daily life, 3) Mood symptoms, and 4) Involvement in a research study. Participant suggestions for app features are listed where relevant.

**Drivers of engagement.** Drivers of engagement were factors directly associated with use of, or disengagement from, the PolarUs intervention. This includes participant perceptions of app features and content, and their experience of tasks

**Table 1. Participant demographics.**

| ID | Age | Gender | Diagnosis | Ethnicity | Location | Employment | Education | Completion of daily check-ins | Completion of weekly check-ins |
|----|-----|--------|-----------|-----------|----------|------------|-----------|-------------------------------|--------------------------------|
| 1 | 34 | Woman | BD-I | White/European | U.S. | Full-time | Undergraduate | 86% | 92% |
| 2 | 22 | Woman | BD-I | East Asian | Canada | Full-time | Highschool | 94% | 85% |
| 3 | 40 | Man | BD-I | White/European | U.S. | Full-time | Master's | 86% | 92% |
| 4 | 35 | Man | BD-I | White/European | U.S. | Student | Undergraduate | 64% | 62% |
| 5 | 55 | Woman | BD-II | White/European | Canada | Disability | Certificate/Diploma | 56% | 85% |
| 6 | 28 | Man | BD-I | White/European | Canada | Full-time | Certificate/Diploma | 77% | 85% |
| 7 | 33 | Man | BD-I | Middle Eastern/ South Asian | U.S. | Student | Master's | 100% | 77% |
| 8 | 21 | Non-binary | BD-I | White/European | U.S. | Student | Certificate/Diploma | 86% | 85% |
| 9 | 49 | Woman | BD-I | White/European | Canada | Full-time | Undergraduate | 41% | 46% |
| 10 | 49 | Woman | BD-I | White/European | Canada | Part-time | Master's | 100% | 100% |
| 11 | 40 | Woman | BD-I | South Asian | Canada | Full-time | PhD | 95% | 77% |
| 12 | 38 | Woman | BD-I | White/European | Canada | Volunteer | Highschool | 98% | 100% |
| 13 | 44 | Woman | BD-I | White/European | Canada | Part-time | Undergraduate | 77% | 100% |
| 14 | 60 | Woman | BD-I | Indigenous | Canada | Disability | Certificate/Diploma | 94% | 100% |
| 15 | 20 | Woman | BD-NOS | East Asian | Canada | Student | Undergraduate (incomplete) | 23% | 31% |
| 16 | 51 | Woman | BD-I | White/European | Canada | Full-time | Certificate/Diploma | 45% | 62% |
| 17 | 40 | Woman | BD-I | White/European | Canada | Full-time | Undergraduate | 76% | 100% |
| 18 | 48 | Woman | BD-I | White/European | U.S. | Disability | Certificate/Diploma | 100% | 92% |
| 19 | 38 | Man | BD-I | South Asian | Canada | Full-time | Certificate/Diploma | 96% | 85% |
| 20 | 27 | Woman | BD-I | Hispanic | U.S. | Full-time | Undergraduate (incomplete) | 52% | 54% |
| 21 | 58 | Woman | BD-I | White/European | U.S. | Part-time | Master's | 100% | 100% |
| 22 | 40 | Man | BD-I | White/European | U.S. | Full-time | Master's | 100% | 100% |
| 23 | 34 | Man | BD-I | White/European | U.S. | Full-time | Master's | 60% | 62% |
| 24 | 31 | Non-binary | BD-I | White/European | Canada | Part time | Undergraduate | 97% | 100% |
| 25 | 23 | Non-binary | BD-I | White/Latin American | U.S. | Disability | Highschool | 13% | 23% |

associated with the intervention. Prominent drivers of engagement were 1) Motivations, 2) Salience, and 3) Perceived effort.

**Motivations:** A common motivation to use the app was to improve symptom management and wellbeing. For some, this motivation was derived from an expectation that the app would be helpful, or observations of beneficial impacts. Confidence that the information contained in the app would be helpful was based on the knowledge that the contents were vetted by the research team for scientific credibility: "*If it has been studied, that it should be able to work for me too.*" [Participant 25] Evidence of improvement could be obtained by self-monitoring, which reinforced engagement. One participant said of reviewing their results: "*I can see how much I've accomplished so maybe I want to do more and like accomplish more of it? It's kind of like a little game*". [Participant 2] However, participants who did not experience benefits or did not expect to receive benefits described less motivation to use the PolarUs app. For example, one participant explaining periods of disengagement noted "*I could track my moods, but I don't know how tracking that's gonna benefit me.*" [Participant 15].

Motivation could be derived from interest in the PolarUs app and its content. For example, one participant described intense initial engagement due to interest in the information and resources: "*I did find it interesting reading through it like when I first got the app I read through every single thing.*" [Participant 7] Other individuals were already familiar with the

psychoeducational materials, and consequently reported lower engagement: "*It wasn't my go-to spot for any information about the illness…I've read a lot about the disorder so I'm pretty well versed.*" [Participant 10] Interest waned over time, particularly due to the repetitiveness of daily check-ins or available resources: "*It seems like the resources never change… that kind of makes them…less intriguing, less inviting*". [Participant 22].

To increase motivation, some participants suggested use of novelty and gamification. Requested game-playing elements included awards, streak counters, and celebratory feedback. One participant suggested gamification elements as expected benefits, while motivating, were not always sufficient to encourage use: "*It's hard for me to get to the point of being like 'okay, I know it will be good for me, so I'm gonna do it.*" [Participant 25]. Novelty was requested in the form of content updates and variation in the daily questions.

**Salience:** To engage with the PolarUs app, participants needed to be aware of it. Notifications amplified the salience of the app, and prompted users to complete their daily check-in. When a technical issue impacted the delivery of daily notifications for a brief period, a number of participants reported that they forgot to use the app, despite having previously consistent engagement: "*While the push notifications were coming, I was engaging with the daily, you know, the daily and the weekly check-ins and things and then they stopped arriving and I just kind of got out of it.*" [Participant 4]. Additional or alternate forms of notifications (e.g., emails, calendar appointments) were suggested by some as a way to support engagement.

Although daily notifications could encourage completion of the check-in items, some participants noted that the psychoeducational materials were less salient. For example, one participant described the content as "*Out of sight, out of mind.*" [Participant 23] Another noted that the app offered little in the way of reminders or encouragement to engage with strategies: "*Just from the interface the entire focus is on do the daily or the weekly and the rest all kind of falls aside.*" [Participant 22] Participants requested additional self-monitoring items or notifications related to their chosen strategies to prompt them to engage with this content, or to implement strategies. For example, one participant suggested "*I need to get kind of a daily something, a reminder, or a daily, I don't know, check in, checklist, something…Just to put it back in my head.*" [Participant 10].

**Perceived effort:** The perceived effort associated with use of the PolarUs app could influence willingness to engage with various features. The brevity of the daily check-in was seen as relatively low effort, which was sometimes contrasted to previous experiences of more burdensome forms self-monitoring. For example, one participant reported not using other mood tracking apps due to the effort involved: "*I don't think the amount of work that I'll need to put into it will be equal to what I'll get out of it…so I think mostly that it, that the [PolarUs] app not going into great detail is a positive.*" [Participant 21] While the daily check-in was described as manageable, a few participants described the longer weekly and monthly QoL surveys as a barrier to engagement: "*There's the really long 50 question ones that are less frequent, but those were kind of like "ugh, I don't want to do that". So, that was one of the reasons why I wouldn't go back into the app.*" [Participant 25].

Reading and engaging with self-management materials in the PolarUs app was comparatively more effortful. Participants attributed this to the length and amount of content, for example, one participant stated: "*Sometimes I found it a bit overwhelming, like a lot of them were, like a lot of reading or a like a lot of like lists where it was like '100 things that could help you in this moment' and I was like 100 is way too many for me to look through and feel like I can absorb right now.*" [Participant 24] Other participants described the number of tasks involved in accessing self-management information and resources as excessive: "*That's one too many steps…because not everybody would have the patience to go inside and dig for that information.*" [Participant 7].

To address burdens associated with PolarUs content, participants suggested the provision of additional structure and alternative ways to engage. Some users expressed a desire for key messages, recommended starting points and next steps that could support their implementation of strategies: "*To start small with like one little thing and then like maybe have a couple of steps…some of them felt kind of impossible to me, because it was just like a huge like concept.*"

[Participant 24] Participants asked for audiovisual content, stating that this was more engaging or would allow them to learn while doing other activities.

Of note, perceived effort associated with the PolarUs intervention may be a particularly critical aspect of engagement, due to its intersecting influence with other drivers. For example, one participant described initial disengagement due to the perceived effort of app use as contributing to an eventual loss of salience: "*I don't want to deal with like you know, the messaging is going to pop up I have to deal with that. I think it was just you know it got to the point where I wasn't thinking about it as much.*" [Participant 23] In another account, motivation to use the app was outweighed by the perceived effort: "*I would love to track like over an extended period of time, like how my moods are fluctuating. But like I just couldn't get myself to keep doing the questionnaires.*" [Participant 25].

**Contextual influences.** Contextual influences had an effect on use of the PolarUs intervention by altering or amplifying a particular driver of engagement. A single contextual influence could have effects on multiple drivers of engagement, and could enhance or detract from intervention use. Prominent contextual influences included 1) The smartphone ecosystem, 2) Competing demands of daily life, 3) Mood symptoms, and 4) Involvement in a research study.

**The smartphone ecosystem:** Some participants noted that the PolarUs app was competing for their attention with other apps. Social media and online shopping were as powerful distractors, and as described by one participant, the PolarUs app "*wouldn't even come into my radar right now*". [Participant 9] PolarUs app notifications were drowned out by other notifications, which could contribute to feelings of overwhelm: "*I have so many notifications sometimes that they just like it's so much for me and I'm like 'I can't do this all right now.'*" [Participant 8]. Some users devised strategies to support the salience of the PolarUs app, such as setting additional reminders, or positioning the app icon alongside frequently used apps. For example, one participant stated: "*I put the PolarUs icon next to the game that whenever I go to open the game, which is gonna happen close to within the first 30 minutes I'm awake, I see it, and then I'm like 'oh well, that's a positive experience' and so I just click on it.*" [Participant 21].

**Competing demands of daily life:** A number of participants noted more salient or higher priority tasks such as school, work, and caring duties often took precedence over check-in completion, or limited the depth with which participants could engage with the content. Several participants noted difficulty finding times where they were consistently available to complete their check-in, meaning notifications often occurred while participants were busy with other activities: "*Even with the reminder, sometimes I wouldn't 'cause it's like you look at the reminder, but maybe you're doing something else and you just forget about it.*" [Participant 16].

**Mood symptoms:** BD mood symptoms were reported to influence engagement with the PolarUs app. Depressive symptoms were linked to a reduction in energy and interest, with consequent negative impacts on drivers of engagement. Participants reported reduced motivation: "*I'm in quite a bad depression, activity-wise I am just not interested in anything or don't do things that are important, so I think that's part of it too.*" [Participant 14] While some participants reported that the brief two item daily check-in was feasible to complete even when depressed, others expressed that the perceived effort of engaging with the PolarUs app was amplified: "*When I was having more depressive symptoms or lower symptoms, it was more work to go on it just because everything is, even the basics.*" [Participant 13] It was also more difficult for the app to achieve salience: "*It would be hard for me to even think about the app, even if I had notifications.*" [Participant 8].

To encourage engagement during depression, interviewees suggested modifications to the PolarUs app. Participants emphasised the importance of simplicity during times when they had less energy. More proactive support from the app was desired, including follow-up when participants failed to complete their daily check-in, and suggestions for relevant resources. For example, one user requested that the app redirect their strategies to something that could specifically support with depressive symptoms: "*Switch my priorities to survival priorities…Like if I was just in a shitshow state, I would just want straight up resources of how to be okay, as much as possible…I don't want to fix my finances*" [Participant 1] Beyond functionality, participants suggested that the tone of communication from the app was important: that it be positive and encouraging, such as the use of affirmational messages.

   

Fewer participants spoke to the influence of manic symptoms on engagement. In these accounts, hypomania and mania altered interest in the PolarUs app; often, priorities shifted to more engaging activities. One participant described difficulties using the app at a time when they were more focused on socialising: "*I had a manic episode and I kinda went back to my old habits…the last thing on my mind was using the app.*" [Participant 20] Some participants avoided smartphone use during periods where they were recovering from mood elevation.

While many participants described disengaging as a result of mood symptoms, a comparable subset of participants described how mood symptoms encouraged use of the PolarUs app, as they sought out coping resources. For example, one participant described responding to manic symptoms: "*I went in and looked around for more resources like as a cry for help. To sort of see what was available.*" [Participant 17].

**Involvement in a research study:** A number of participants situated their engagement with the PolarUs app in the context of the research study. Often, use of the PolarUs app was driven by a desire to be helpful, both for the research team, as well as other people living with BD who may benefit from the app. Reflecting on their use, one participant described a sense of responsibility to provide sufficient data: "*I mean the part that got me hyped up was that – my feedback would provide a way for – now I'm feeling guilty for not using it more – would provide information that would help it be a valuable tool for other people.*" [Participant 14] Other participants noted that the study amplified the salience of the app. For example, one participant described re-engaging in the period approaching an assessment interview: "*Knowing that I had a new interview coming up would remind me…I'd go 'oh no I haven't been checking in', and so I just keep doing it again.*" [Participant 5] Some participants questioned the degree to which they would have engaged outside of this context: "*I wanted to keep using it because I wanted to be helpful…Whereas I probably would have bailed earlier.*" [Participant 13].

## Discussion

The aim of this study was to explore experiences of engaging with a self-management app for BD, in the context of a pilot trial of the PolarUs app. Findings offer considerations for clinicians and researchers seeking to optimise engagement in digital self-management interventions in BD, both in regards to optimal app design and out-of-app supports.

Aspects of these qualitative findings broadly align with existing theories characterising how individuals engage with technology, such as the widely used Technology Acceptance Model (TAM; [21]), an extension of the earlier Theory of Reasoned Action developed to explain adoption of health behaviours [22,23]. The TAM emphasises the role of perceived usefulness and ease of use of a system in explaining how and when people use a technology [21], which is echoed by our qualitative findings highlighting the importance of motivations and perceived effort in engagement with the PolarUs app. Present findings elaborate on this theory in the context of how individuals with BD use self-management technologies, such as the types of motivations expressed and features that contribute to perceived effort. For example, we found that motivations could extend beyond perceived usefulness, and included seeking experiences of enjoyment and curiosity. Findings also revealed that perceived ease of use could vary between app components, as daily check-ins were construed as less effortful than navigating, reading, and applying psychoeducational content. Our analysis suggest that app salience may be a precursor to engagement, and could help explain findings that positive perceptions of usability and therapeutic effectiveness may not correlate with app use [24], but this relationship would need exploration in future analyses. One possibility is to conduct structural equation modelling, which has been applied in non-health digital platforms to model interrelationships between subjective and behavioural aspects of engagement [25,26]. Future research in BD could operationalise the constructs of salience, motivation, and perceived effort using diverse behavioural and self-report data, to examine whether the reciprocal influences described by participants in this qualitative study can be demonstrated quantitatively.

The drivers of engagement highlighted in this qualitative study also map onto other frameworks, including the Capability, Opportunity, Motivation, and Behaviour model (COM-B) [27], and the Theoretical Domains Framework (TDF) [28], Both of

these models have been used to classify factors associated with app engagement [29]. For example, the drivers of perceived effort and salience identified here align with the COM-B, which highlights the importance of psychological capabilities, and the TDF, which further specifies this as including memory, attention, and decision-making processes. Similar app design choices to those commented on by participants in this qualitative study have been reported to act on these aspects of the COM-B and TDF, including well designed reminders and navigation supports [29]. The driver of motivation identified here also maps onto both the COM-B (as 'reflexive' and 'automatic motivation') and TDF (as 'beliefs about consequences'). App features found to support motivation in the broader literature once again parallel participants' desires for gamification and feedback about users' progress [29]. We note that not all aspects of the COM-B and TDF were represented in participant feedback, especially those influencing uptake of an app in the first instance (e.g., app literacy, availability, and accessibility). Given that we interviewed current users of the PolarUs app, this suggests a need to also consult with non-users to explore if any of these factors would influence their willingness to commence use of a BD self-management app.

Participants reported a range of contextual influences on their engagement with the PolarUs app, including the competing demands of daily life, the smartphone ecosystem, the research trial context, and the experience of BD-related mood fluctuations. Taken together, these qualitative findings reiterate the complexity of assessing and explaining engagement with a mental health app, especially in light of the heterogenous and episodic nature of BD [12]. Although the models of engagement discussed here broadly consider aspects of the individual, the intervention, and the context that may affect engagement, our findings suggest a need to consider how these models operate in the specific population of interest. For example, the COM-B and TDF have considered the role of social influences (e.g., health practitioner support, interaction with peers), which may go towards explaining the influence of the research trial context, where contact with trial staff was experienced as both motivating and served as a reminder for app use. However, other contextual influences discussed this present paper are not fully addressed in these models, such as the role of fluctuating mood symptoms and competing attentional demands in the environment. While we do not set out to propose a new model of engagement or assessment framework, our findings highlight in future feasibility research considering the user experience of people living with BD, consideration of diverse intervention-related, individual, situational and dynamic factors, is necessary to make sense of patterns of engagement with mHealth interventions.

A number of features were proposed to improve engagement with the PolarUs app. Gamification was suggested as a means to encourage motivation to use the PolarUs app, and may include game elements such as levels or progress feedback, points or scoring, rewards or prizes, narratives, and personalisation [15]. The potential of gamification for enhancing engagement has been demonstrated in a head-to-head comparison of two versions of a smartphone app for anxiety [30]. However, whether or not gamification promotes clinically meaningful engagement [31], or simply "engagement for engagement's sake" [32], is debatable. A review of gamification elements in mental health apps suggested that their inclusion was primarily chosen to enhance intervention engagement; fewer studies have considered if and how gamification could enhance an intervention's efficacy [15]. In addition, the utility of these elements for mood disorders apps may require further examination, as one meta-analysis of gamified apps for depression found no impact on adherence or efficacy [33]. Decreased reward sensitivity may decrease motivation to obtain rewards during depression. Our own survey of app feature preferences held by people with BD identified concerns that gamification could encourage excessive reward-seeking, which may increase the risk of problematic mood elevation [34]. Indeed, some participants in this study reporting avoiding smartphone use during periods where they were recovering from hypomania/mania, potentially reflecting such concerns about excessive reward seeking. As such, while gamification may be one strategy to encourage motivation, its application in BD apps must be evaluated in terms of engagement, efficacy, and unwanted or adverse effects.

Participants in this study made suggestions about how the app could support their exploration of the content through a guided, scaffolded approach. Although there is some research to suggest that engagement in web-based interventions is enhanced by allowing users free choice over when and how they interact with content [35], our findings suggest that this may increase perceived effort. Participants in an evaluation of a web-based mindfulness intervention similarly reported

that simultaneous release of all intervention content was overwhelming [36,37]. Personalising the delivery according to user characteristics and self-monitoring information, known as digital phenotyping, may avoid overwhelming users with too much choice/information [38]. This approach has been validated by collecting passively sensed data to create predictive models of depressive symptoms [39]; user attitudes about the utility of the app were associated with rates of completion of recommended tasks related to the passive data collected.

The experiences of PolarUs users draws attention to the potential of out-of-app therapeutic techniques to support consistent, clinically impactful app use. For example, although originally developed to treat substance use, motivational interviewing can be applied to encourage the adoption of diverse health behaviours, and aims to resolve ambivalence to change by exploring the downsides of the status quo, identifying benefits of change, and building positive expectancies of change [40]. Motivation-enhancing strategies may be particularly relevant in the period before people start to experience the benefits of an mHealth intervention, at which point app use becomes more self-sustaining [34,41]. Clinicians may also play a role in reducing the effort associated with navigating complex interventions. While advanced algorithms may eventually aid in the personalized delivery of app content, digital phenotyping in BD is at a relatively nascent stage of development [42]. In the interim, clinicians may use their knowledge of case formulation and evidence-based therapeutic approaches for BD to make recommendations about what app features and content a user would benefit from. Strategies used to support individuals with medication adherence may also be relevant, given both app and medication use could be impacted by difficulties with attention, executive function, and memory. These cognitive challenges are common in BD [43], and are thought to impact adherence to medications [44–46]. Compensatory strategies that can encourage use of medications as prescribed (e.g., external reminders and association strategies) may also be helpful for supporting individuals with BD to remember to engage with apps on a routine basis [47,48].

Clinicians may also play an important role in supporting engagement during periods where patients are experiencing mood symptoms. For instance, clinicians may wish to consider offering additional reminders and encouragement at times when patients are experiencing increased depressive symptoms, as both present findings and earlier qualitative research suggest decreased motivation and energy likely impact app engagement [49]. Techniques drawn from cognitive behaviour therapy, such as activity scheduling and problem-solving may be useful to overcome practical and motivation-based barriers to app use [50]. Some people with BD may increase their use of digital mental health tools during depressive periods: here and in other research, this appeared to be driven by the perception that these interventions provided valuable support [51]. Clinicians may encourage this by reviewing self-monitoring data from periods where app use was more consistent, as seeing evidence of benefits was described as motivating. Limited existing literature suggests that during periods of euthymia, people with unipolar depression report being more willing to engage with self-management apps on account of their relatively higher energy levels [49]. However, while increased use of mental health apps tends to support mental health outcomes in studies of unipolar depression [52–54], clinicians working with individuals with BD may need to be sensitive to times where increased engagement is may carry risks. As noted earlier, some instances of avoiding smartphone use were reported in this study, a finding which has been echoed in other explorations of technology use in BD [34,55]. Further exploration of what optimal engagement looks like in the context of the episodic and heterogenous course of BD is necessary to best support patients to use mHealth in a clinically effective and safe manner [12].

## Method

The present qualitative study is embedded in a quantitative, pilot evaluation of the feasibility and impact of the PolarUs smartphone app [20]. In the overarching study, individuals with a confirmed diagnosis of BD were provided with access to the PolarUs smartphone app for a three-month period. An embedded mixed-methods qualitative research design was used [56]: a purposive subsample of study participants was interviewed to obtain in-depth feedback on experiences using the PolarUs app, including behaviours, impacts, and intervention engagement. Here, the focus is on reporting salient aspects of user experience of engaging with the PolarUs app over a three-month intervention period.

The qualitative design was phenomenological in nature, using thematic analysis to explore the subjective experiences and perspectives of individuals who had used the PolarUs app [57]. The analysis is reported according to the Standards for Reporting Qualitative Research Framework (SRQR) [58], in addition to recommendations to ensure quality practice in thematic analysis [59]. A copy of the SRQR checklist is available in S1 File.

Approval for the study was granted by the University of British Columbia's Behaviour Research Ethics Board (H21-02042). The authors assert that this work complies with the ethical standards of the relevant national and institutional committees on human experimentation and with the Helsinki Declaration of 1975, as revised in 2008. All participants received written information on the study and gave written consent to be contacted for an interview. Data in the study was treated confidentially and transcripts de-identified. Interview participants received a gift certiciate honorarium ($30 CAD/$25 USD).

## The PolarUs App intervention

The PolarUs app was developed by the Collaborative Research Team to study psychosocial issues in Bipolar Disorder (CREST.BD), and provides psychoeducation and self-monitoring on aspects of QoL relevant to BD [60]. The PolarUs app incorporates and expands on material from a previously evaluated CREST.BD web-based self-management intervention (www.bdwellness.com), which was found to have positive impacts for subjective recovery and QoL [61]. Self-monitoring was enabled by the inclusion of a widely-used BD-specific QoL self-assessment tool, the QoL.BD [62,63], a digital adaption of which has been previously subject to psychometric evaluation [64].

The user flow is presented in full detail in the protocol [20]. In brief, the PolarUs app allows users to freely explore psychoeducation content grouped around 17 different QoL domains assessed by the QoL.BD (mood, sleep, cognition, home management, identity, independence, leisure, finances, relationships, spirituality, exercise, diet, sexual health, substance use, self-esteem, work and study). Each domain summary contains evidence-informed strategies that the user can apply to self-manage the impacts of BD on this life domain. After an initial QoL self-assessment using the QoL.BD, users are free to select up to three life domains to focus on, and a further four self-management strategies to implement over the following month (see Fig 1). Psychoeducational content is provided to help users understand the relevance of each QoL domain and self-management strategy. Additional practical resources are provided to support users in implementing the strategies (e.g., worksheets, videos, blog posts, and websites). Users receive encouragement to use the app in the form of a daily notification, co-authored by individuals with lived experience. In-app data was collected in the form of two daily self-report items assessing mood and sleep (collectively referred to as the 'daily check-in'), weekly self-assessment using the brief 12 item version of the QoL.BD (referred to as the 'weekly check-in'), and monthly completion of the full 56 item QoL.BD (see Fig 2).

## Participants

Participants in the qualitative study were a subsample of those enrolled in the overarching evaluation study ($n = 170$) who provided consent to be contacted regarding participation in a follow-up telephone interview ($n = 156$).

**Inclusion/exclusion criteria.** Participants in the overarching evaluation study were required to be: i) aged 18 years or above, ii) able to communicate in English, iii) able to provide informed consent, iv) a resident of Canada or the United States, v) a smartphone user, and vi) diagnosed with BD (BD-I, BD-II, or BD not otherwise specified) as confirmed by a structured clinical interview at baseline [65]. Individuals experiencing psychosis or significant suicidal ideation at the time of the diagnostic interview were excluded.

In line with SRQR standards [58], purposive sampling [66] was used to select participants to invite into the qualitative study. Potential participants were invited on the basis of demographic criteria to promote diversity in the sample with respect to age, gender, BD diagnosis, ethnicity, education, employment, and location (Canada/United States). In addition, we invited people who demonstrated varying degrees of engagement with the app (operationalised by the proportion of

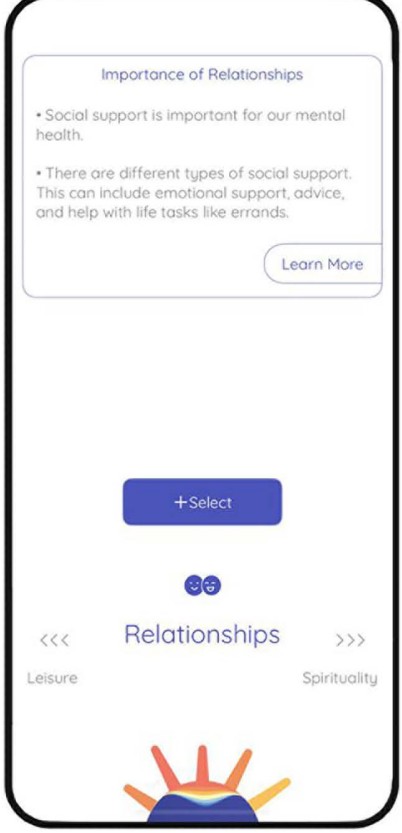
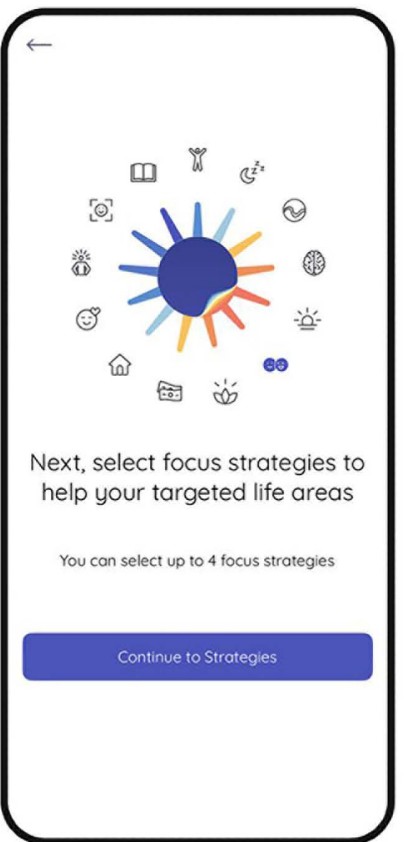
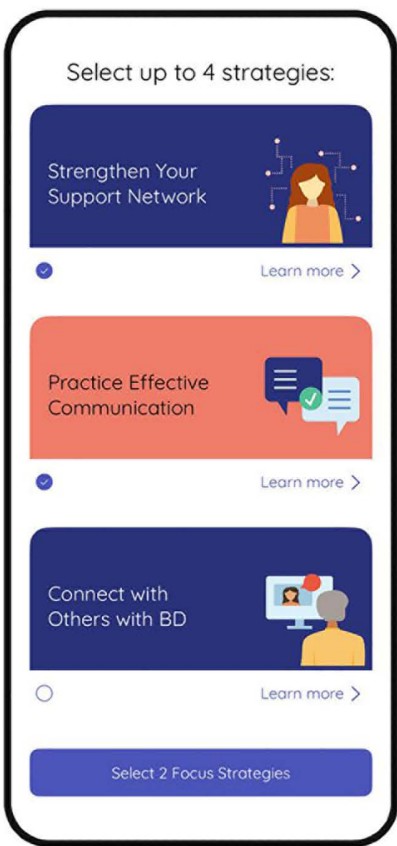

**Fig 1. Selection of focus life domains and self-management strategies in the PolarUs app.**

total possible in-app daily and weekly check-ins completed during the study period). The approach to purposive sampling was theoretical; invited candidates were selected on the basis of diversity achieved in the interviewed sample to date [67,68].

As the positivist assumptions of data saturation are not coherent with the reflexive nature of thematic analysis and do not safeguard against thin or underdeveloped themes [69], our final determination of sample size was guided by information power [70]. An initial target sample size of approximately 30 individuals was based on the narrowness of our study aim relative to larger qualitative evaluations of digital interventions with more components and greater between-individual variation [61,71]. This initial target sample was intended to be flexible, recognising that appraisal of information power should be repeated during data collection to allow the quality of dialogue and purposive sampling aims to influence final recruitment decisions [70]. Given that the majority of interviews were highly rich and detailed, and to avoid further weighting the sample to towards high engagers from over-represented demographics (i.e., White women), the research team agreed to conclude recruitment once 25 participants were interviewed.

**Recruitment.** Recruitment into the overarching study was conducted via social media posts (Facebook, Twitter, and Instagram), the CREST.BD website and mailing list, and outreach via partner organisations. At enrolment, individuals indicated their consent to be contacted at a later date regarding participation in a follow-up interview.

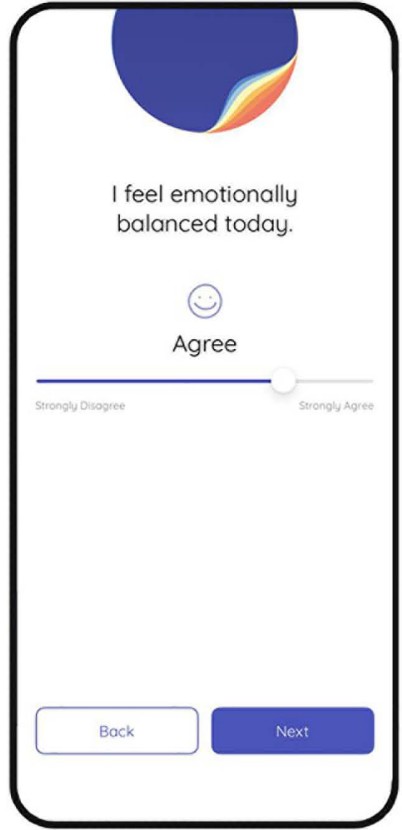
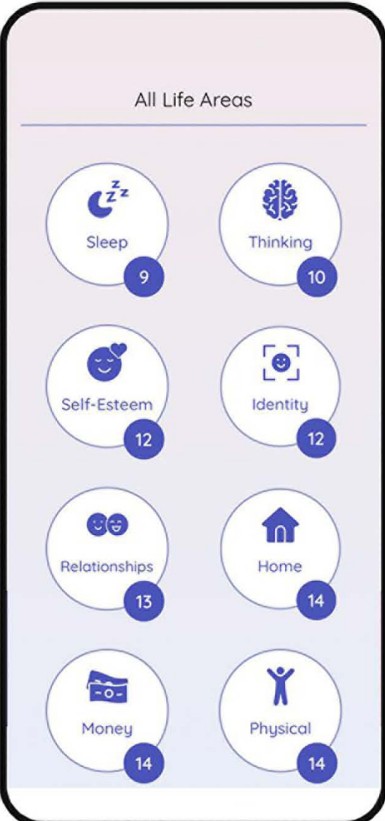
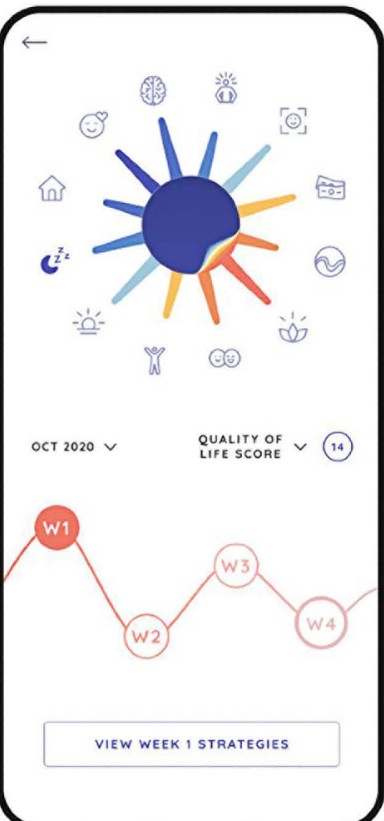

**Fig 2. Self-assessment features in the PolarUs app.**

## Qualitative interview

A semi-structured interview schedule was developed with three sets of open-ended questions (S2 File). The first set of questions asked participants about their perspectives of the app, including specific likes, dislikes, and comparisons to other health apps they had used. The second set of questions explored experiences of engagement and disengagement with the app, as well as factors that influenced this. Finally, participants were asked about their experiences of implementing self-management strategies. Probes and reflective listening were used to elicit in-depth responses.

## Procedure

Participants in the overarching study who provided consent to be contacted were emailed after completion of their final clinician-administered assessment, given written information on the qualitative interview, and invited to indicate their consent and schedule an interview time. Interviews were conducted by the first author (EM) using the Zoom teleconferencing platform, digitally recorded and transcribed verbatim.

## Data analysis

The first author (EM) followed Braun and Clark's [57] guidelines for thematic analysis. Data familiarisation occurred through reading and re-reading transcripts. Data were assigned brief descriptive codes in NVivo 12 (QSR International,

2018). An inductive coding approach was used; codes were guided by participant responses, rather than pre-conceived theories or ideas. This approach was selected given the dearth of prior literature on engagement with mHealth interventions for people living with BD, as it allows for the development of themes that represent the unique contexts and experiences of participants, including topics that may not have been apparent when data is viewed through the lens of an existing framework [57,72]. Codes were reviewed and interpreted by the research team, and themes were generated to describe patterns in coding. Provisional themes were iteratively reviewed during development for coherency, meaningfulness, richness, and relationships to other themes. The essence of the most important themes in relation to the study aim are described herein with illustrative quotes.

Multiple techniques were used to support analytic rigor and trustworthiness, leveraging the widely-used criteria developed by Lincoln and Guba [73], as well as specific recommendations in the SRQR [58]. In accordance with thematic analysis standards and to support reflexivity [59,74], we note that interviews and data analysis were led by the first author EM, a white woman, psychologist and mid-career researcher with expertise in QoL-focused interventions for BD and digital health intervention. Our use of a single primary analyst is in line with quality practices in reflexive thematic analysis; positivist 'coding reliability' approaches are inconsistent with the philosophy and values of this approach, where knowledge generation is situated as a contextual and interpretative process [59]. Rather, to support credibility (the alignment between participant views and the researchers' representations of these), peer debriefing was engaged in throughout the analytic process: co-authors EEM and RH reviewed and discussed the descriptive accounts of themes and transcripts for coherency and validity of interpretation, and regular analytic meetings were held with co-authors EEM, RH, and HOB to discuss coding, themes and their organisation, and the selection and reporting of themes. Credibility was also upheld through member checking conducted with the PolarUs User Group (PUG) [66,75,76], consisting of individuals with lived experience of BD, many of whom participated in the overarching mixed-methods study and qualitative interview. No PUG members expressed dissatisfaction with the theme descriptions. Negative case sampling was achieved by deliberately recruiting individuals who displayed low engagement with the PolarUs app, and representing the granularity of experiences of engagement. Other strategies applied included the use of thick description of the research setting (i.e., the PolarUs app evaluation study) and participant group to ground the transferability of findings (their potential to be applied in other settings). Dependability (consistency of research processes) was supported through documentation of the study methodology, intervention, interview guide, and qualitative approaches used to guide thematic analysis. Confirmability (the degree to which researcher interpretations are grounded in data) was supported through use of participant quotes.

## Limitations

Limitations to the present design should be noted. First, although we aimed to include individuals with varying levels of app engagement, most of the interview participants were highly engaged. Hence, perspectives from those with lower app use were limited. Further, findings may only reflect the views of those to individuals willing to engage in research, as evidence by participant descriptions of the contextual influence of the study. This finding accords with the hypothesis that aspects of trial design (e.g., contact from research assistants) may encourage greater engagement [77]. In a systematic comparison, median rates of engagement in clinical trials were four times higher than real-world usage of the same program. Our data suggests that the trial setting may increase altruistic motivations to use the app, as well as increase app salience. However, as we did not contact individuals who formally withdrew from the study, we are not able to represent the perspectives of individuals who may have found aspects of the research study itself aversive or burdensome. Evaluations of real-world usage metrics are important, as findings from the feasibility evaluation may or may not generalise outside the trial setting.

Second, although generalisability of findings is not a key aim of qualitative research [78], over-representation of particular demographic groups in the sample should be kept in mind when interpreting the experiences reported. Despite attempts to invite a broad range of participants, the sample was predominantly White, highly educated, women. Given

that access to, familiarity with, and acceptability of mHealth may be impacted by gender, ethnicity, and education [79–81], future evaluations of PolarUs in more diverse populations is needed to explore any potential impacts of the digital divide on engagement [82].

Third, we chose to contact participants after the conclusion of the three-month intervention period, in order to avoid prematurely labelling participants as withdrawn given that app users may disengage and re-engage over time [12]. Contacting participants after this time point may have contributed to low response rates, especially for individuals who disengaged early in the study. The duration of time between the completion of the intervention period and qualitative interviews was also influenced by scheduling difficulties (in particular, over the December/January holiday period). Taken together, this delay may have impacted the ability of individuals to recall factors that influenced their engagement over the course of the study. Future studies could balance the risk of recall bias for early disengagers with the need to represent the influence of various drivers of engagement/contextual influences over time by conducting repeated qualitative interviews. This approach was demonstrated in a study that conducted weekly interviews with individuals with depression who used two smartphone apps [49]. Repeated interviews generated insights into the fluctuating nature of depressive symptoms and how apps could respond to periods of low energy and motivation, as well as leverage euthymic periods to establish healthy routines, and proactively plan for future depressive episodes. Using this approach in future research may yield important insights about the dynamic use of digital health tools in the context of the chronic, relapsing course of BD. However, some factors reported here to influence app engagement (life circumstances, ongoing symptoms) are also likely to impact qualitative interview scheduling, and the risk of impacted recall must be balanced with including individuals who may face more challenges to engagement and therefore have highly relevant insights to offer. As such, in this study, no restrictions were based on time elapsed between the conclusion of the intervention period and the qualitative interview. We further note that the optimal recall period for self-report surveys is suggested to be 6 months or less [83], as such, even for participants who disengaged use of the PolarUs app early, the timeframe of our interviews would still be well within recommendations. Finally, the time elapsed since the intervention in our analysis also contributed to our ability to find that the research context was a unique contributor to patterns of engagement with the PolarUs app, as some participants commented on the difference in their use post-trial as compared to when the study was ongoing. In light of ongoing discussions about how best to measure engagement with mental health apps, and concerns regarding the potential for the study environment to positively bias engagement as compared to real-world contexts [9,12,77], we argue that despite the potential of impacts to recall ability, these interviews represent an important contribution to the broader mHealth literature.

## Conclusions

This qualitative investigation of user experiences of the PolarUs self-management app identified that engagement was influenced by a range of factors, including app features, user attitudes towards the app, and contextual influences. Findings highlight a tension between enhancing app usability and promoting engagement with features and content likely to be clinically impactful, as well as the importance of looking beyond behavioural metrics and usability ratings in evaluating the feasibility of app-based interventions. Of use to both healthcare providers and researchers, this exploration has yielded insights into aspects of app design and out-of-app supports which may strengthen clinically meaningful engagement with digital self-management interventions in a BD population.

## Supporting information

**S1 File. Standards for reporting qualitative research framework checklist.**
(DOCX)

**S2 File. Qualitative interview guide.**
(DOCX)

## Acknowledgments

The authors gratefully acknowledge the research participants who were involved in this project. We thank the CREST.BD network members, advisory groups, trainees, volunteers, and research staff for their contributions to the development of the PolarUs app and study design.

## Author contributions

**Conceptualization:** Emma Morton, Erin E. Michalak.

**Formal analysis:** Emma Morton, Rachelle Hole, Erin E. Michalak.

**Funding acquisition:** Emma Morton, Rachelle Hole, Heather O'Brien, Linda Li, Steven J. Barnes, Erin E. Michalak.

**Investigation:** Emma Morton.

**Methodology:** Emma Morton.

**Writing – original draft:** Emma Morton.

**Writing – review & editing:** Rachelle Hole, Heather O'Brien, Linda Li, Steven J. Barnes, Erin E. Michalak.

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
