## [Decision Letter · Decision Letter 0]

30 Oct 2024

PDIG-D-24-00352What influences engagement with a bipolar disorder self-management app? A qualitative investigation of use of the PolarUs app.PLOS Digital Health Dear Dr. Michalak, Thank you for submitting your manuscript to PLOS Digital Health. After careful consideration, we feel that it has merit but does not fully meet PLOS Digital Health's publication criteria as it currently stands. Therefore, we invite you to submit a revised version of the manuscript that addresses the points raised during the review process. Please submit your revised manuscript within 60 days Dec 29 2024 11:59PM. If you will need more time than this to complete your revisions, please reply to this message or contact the journal office at digitalhealth@plos.org.  Please include the following items when submitting your revised manuscript:* A rebuttal letter that responds to each point raised by the editor and reviewer(s). You should upload this letter as a separate file labeled '?>Response to ReviewersRevised Manuscript with Track ChangesManuscript**Journal Requirements:****Additional Editor Comments (if provided):****Reviewers' Comments:**

**Comments to the Author**

1. Does this manuscript meet PLOS Digital Health’s publication criteria?

Reviewer #1: Yes

Reviewer #2: Yes

Reviewer #3: Partly

2. Has the statistical analysis been performed appropriately and rigorously?

Reviewer #1: N/A

Reviewer #2: Yes

Reviewer #3: No

3. Have the authors made all data underlying the findings in their manuscript fully available (please refer to the Data Availability Statement at the start of the manuscript PDF file)?

Reviewer #1: Yes

Reviewer #2: No

Reviewer #3: No

4. Is the manuscript presented in an intelligible fashion and written in standard English?

Reviewer #1: Yes

Reviewer #2: Yes

Reviewer #3: Yes

Reviewer #1: Dear Authors,

Thank you for sharing this manuscript. I found the detailed qualitative analysis interesting and thorough, it sets a good standard for MH app intervention evaluation. There were a few minor changes I would recommend:

Line 81: Where you reference 'symptoms', which ones? Additionally, for the prevalence statistics - are these global or US based?

Line 131: I would like to see some explanation for the range in interview times either here or in the limitations. Additionally, in the limitations you refer to waiting until the conclusion of the study for the interviews, but is there a reason some of them were so long after this (again, a wide range here).

Method - really great to see member checking of the thematic analysis!

Line 474: I think 'lover' is a typo of 'lower'

I thought this finding was interesting 'Some participants avoided smartphone use during periods where they were recovering from mood elevation.' - it suggests perhaps some associated issues with encouraging online/device engagement in order to use the app, when this could be potentially harmful or interrupt recovery. This should be considered in the discussion or limitations.

Reviewer #2: Recommended decision: Minor review

This study presents the reasons for engagement and disengagement with a bipolar disorder self-management app in a clear and methodologically correct manner. The qualitative focus on engagement provides some new and relevant insights for the development of mental health apps.

Both research question and design are clearly stated and explained. The chosen qualitative interviews and their analysis are clearly stated and suitable for the research question.

Sampling and data collection are clearly described and adequate by using a purposive subsample of participants. What is missing here is a rationale or justification for the chosen sample size of 48 invited patients.

The data analysis process is described in sufficient detail. Strategies to enhance credibility are incorporated by member checking done by individuals with BD experience.

The findings are clearly presented and organized by meaningful and justified themes. However, the comparison of the results with existing literature is lacking; instead the Technology Acceptance Model is used as a comparison which is not sufficient.

Limitations of the study and research contributions are clearly stated and justified.

Reviewer #3: This qualitative study investigated the factors that influenced individuals’ engagement with a bipolar disorder (BD) self-management app. It is an interesting study within the journal scope. However, the research question is not clear, and more details regarding the methods and results should be provided. My detailed comments are as follows.

Overall

- Guidelines and criteria have been developed to ensure the validity and reliability of qualitative research and its analysis. Please ensure that the present qualitative research and its analysis follow these methodology guidelines and criteria. Also, please let readers know about the specific guidelines and criteria that are used. I have considered rejecting this submission due to the unclear validity and reliability of the qualitative analysis. However, I would like to hear what the authors have to say and will evaluate the paper again next time. If the paper still suffers from validity and reliability issues, it will be rejected.

Abstract

- The authors mentioned that they identified the app features associated with the drivers of engagement; however, I was unable to locate these results in the main text. Please organize the findings to improve the clarity.

Introduction

- The research question is not clear. It appears that the authors investigated the factors that influenced engagement with the app, as well as the factors that interacted with drivers of engagement. Please specify if there were two research questions and present them clearly.

Methods

- Please provide more details and figures to illustrate the PolarUs App, as the features are unclear. Although the authors mentioned that details can be found in the protocol, the references to “three life domains” and “4 self-management strategies” are confusing in this manuscript.

- In addition to the interview questions, the authors examined quantitative outcomes such as QoL.BD (both 12-item and 56-item versions). However, the results of these health-related outcomes are missing. The authors should provide the results of the descriptive analysis on these outcomes. In addition, they can examine whether the improvement in health condition can influence engagement with the app.

Results

- It is unclear why the interviews were conducted between 9 and 52 days following the conclusion of the intervention. This broad range might influence the participants’ responses to the interview questions. It is also noted that the mean duration was 28 days, which raised some concerns. The authors need to explain why these interviews were not conducted immediately after the completion of intervention.

- For each of the resulting factors (or outcomes), how many participants mentioned it? Please provide a number for each.

- Along the same lines as the above comment, for “Several participants noted difficulty finding times where they were consistently available to complete their check-in …”, how many is “several”? This comment would have been addressed after addressing the above comment. I see the same issue in other places in the manuscript. Please fix all of them.

- The authors did not specify how to complete daily and weekly check-in. Did they require the participants to log in to the app every day and answer some questions to check in? How did they calculate the percentage of completion? Did they conduct monthly check-in?

- The authors identified three factors that influenced engagement with the app, including motivations, salience, and perceived effort. However, I am concerned whether these factors were derived from existing models or frameworks. Please provide more evidence to support their validity.

Discussion

- Since the authors mentioned the Technology Acceptance Model (TAM), it would be good to know why they did not conduct quantitative analysis related to TAM (e.g., path analysis). Such analysis would provide more important findings regarding the engagement with the BD app.

**Do you want your identity to be public for this peer review?** For information about this choice, including consent withdrawal, please see our Privacy Policy

Reviewer #1: No

Reviewer #2: No

Reviewer #3: No

**Figure resubmission:****Reproducibility:** To enhance the reproducibility of your results, we recommend that authors of applicable studies deposit laboratory protocols in protocols.io, where a protocol can be assigned its own identifier (DOI) such that it can be cited independently in the future. Additionally, PLOS ONE offers an option to publish peer-reviewed clinical study protocols. Read more information on sharing protocols at https://plos.org/protocols?utm_medium=editorial-email&utm_source=authorletters&utm_campaign=protocols

---

## [Decision Letter · Decision Letter 1]

28 Apr 2025

PDIG-D-24-00352R1What influences engagement with a bipolar disorder self-management app? A qualitative investigation of use of the PolarUs app.PLOS Digital Health?

Dear Dr. Michalak,

Thank you for submitting your manuscript to PLOS Digital Health. As with all papers, your manuscript was reviewed by members of the editorial board. Based on our assessment, we have decided that the work does not meet our criteria for publication and will therefore be rejected. If external reviews were secured, reviewers’ comments will be included at the bottom of this email.

We are sorry that we cannot be more positive on this occasion. We very much appreciate your wish to present your work in one of PLOS's Open Access publications. Thank you for your support, and we hope that you will consider PLOS Digital Health for other submissions in the future.

Yours sincerely,

Calvin Or, PhD

Section Editor

Leo Anthony Celi

Editor-in-Chief

PLOS Digital Health

orcid.org/0000-0001-6712-6626

Additional Editor Comments (if provided):

Thank you for submitting the revision. I am sorry that, based on my review, we have decided not to proceed further with your manuscript. I understand that two of the reviewers provided positive comments. However, the third reviewer raised valid concerns that the other two did not identify.

The possibility of recall bias is significant. I can imagine that participants might not fully and accurately recall their experiences after 28.3 days (average) or 52 days (longest) of using an app, particularly if they disengaged early. Simply noting this as a study limitation does not address the issue, as recall bias remains present and could impact the results.

I agree with the reviewer that data saturation should be considered when determining when to stop including additional participants.

I also agree with the reviewer that adhering to a qualitative research reporting guideline is important. You can refer to the following resources:

https://jamanetwork.com/journals/jamasurgery/article-abstract/2778475

https://academic.oup.com/intqhc/article-abstract/19/6/349/1791966

Reviewers' comments:

Reviewer's Responses to Questions

**Comments to the Author**

Reviewer #1: All comments have been addressed

Reviewer #3: (No Response)

publication criteria?

Reviewer #1: Yes

Reviewer #3: (No Response)

3. Has the statistical analysis been performed appropriately and rigorously?

Reviewer #1: N/A

Reviewer #3: (No Response)

4. Have the authors made all data underlying the findings in their manuscript fully available (please refer to the Data Availability Statement at the start of the manuscript PDF file)?

Reviewer #1: Yes

Reviewer #3: (No Response)

5. Is the manuscript presented in an intelligible fashion and written in standard English?

Reviewer #1: Yes

Reviewer #3: (No Response)

Reviewer #1: Thank you for addressing all of my comments.

Reviewer #3: I appreciate the authors’ efforts in revising the manuscript. Some of my comments have been well addressed; however, there are still methodological issues. In general, this manuscript is not yet ready for publication.

- The initial sample size was 30 in the published protocol, while it was reduced to 25 in the current study. The authors concluded recruitment once a sample size of 25 was achieved, which seemed quite casual. They provided justifications that they aimed to “avoid further weighting the same sample towards high engagers from already over-represented demographics (i.e., White woman)”; however, this can be problematic:

1) The issue of over-represented demographics regarding gender and ethnicity can influence the interpretation of the findings. The authors need to discuss the potential limitations.

2) It is unclear whether the other demographics were also over-represented. Stratification by demographics would be recommended.

3) The authors used purposive sampling to promote diversity in the sample, while this method might have introduced selection bias and limited generalizability of the findings.

- Thank you for introducing more details of the inductive approach used in thematic analysis (i.e., data were analyzed without preconceived theories). Although it increased the flexibility of identifying emerging themes and concepts from interviews, I still have some concerns that need further explanation.

1) The inductive approach relies heavily on the researchers’ interpretations, which can introduce bias and affect the credibility of the findings. Without a well-defined coding framework before starting the analysis, how did the authors determine which factors were related to their research questions and whether they should be extracted from the interview transcripts? If the authors did not provide categories, definitions, and examples of coding in advance, how did they ensure consistency among researchers?

2) Although several researchers reviewed and discussed the codes and themes in thematic analysis, it appears that only one researcher (EM) initially read the transcript and performed the coding. To reduce the risk of bias, it is highly recommended that two or more researchers independently transcribe the interviews, read the transcripts, and perform the coding; and then they discuss to resolve any disagreements.

3) Findings from inductive research are often context-specific, making it challenging to generalize results to broader populations or contexts. For example, the authors found drivers of engagement were shaped by contextual influences, including “involvement in a research study”; however, it is hard to use this finding to create robust and applicable insights for future practice.

**Do you want your identity to be public for this peer review?** For information about this choice, including consent withdrawal, please see our Privacy Policy

Reviewer #1: No

Reviewer #3: No

---

## [Decision Letter · Decision Letter 2]

1 Sep 2025

What influences engagement with a bipolar disorder self-management app? A qualitative investigation of use of the PolarUs app.

PDIG-D-24-00352R2

Dear Dr. Michalak,

We're pleased to inform you that your manuscript has been judged scientifically suitable for publication and will be formally accepted for publication once it meets all outstanding technical requirements.

Within one week, you'll receive an e-mail detailing the required amendments. When these have been addressed, you'll receive a formal acceptance letter and your manuscript will be scheduled for publication.

An invoice for payment will follow shortly after the formal acceptance. To ensure an efficient process, please log into Editorial Manager at https://www.editorialmanager.com/pdig/ click the 'Update My Information' link at the top of the page, and double check that your user information is up-to-date. For billing related questions, please contact billing support at https://plos.my.site.com/s/.

Kind regards,

Haleh Ayatollahi

Section Editor

PLOS Digital Health

Additional Editor Comments (optional):

Reviewer #3:

Reviewer #4:

Reviewers' comments:

Reviewer's Responses to Questions

**Comments to the Author**

Reviewer #3: All comments have been addressed

Reviewer #4: All comments have been addressed

publication criteria?

Reviewer #3: Yes

Reviewer #4: Yes

3. Has the statistical analysis been performed appropriately and rigorously?

Reviewer #3: N/A

Reviewer #4: Yes

4. Have the authors made all data underlying the findings in their manuscript fully available (please refer to the Data Availability Statement at the start of the manuscript PDF file)?

Reviewer #3: No

Reviewer #4: Yes

5. Is the manuscript presented in an intelligible fashion and written in standard English?

PLOS Digital Health does not copyedit accepted manuscripts, so the language in submitted articles must be clear, correct, and unambiguous. Any typographical or grammatical errors should be corrected at revision, so please note any specific errors here.

Reviewer #3: Yes

Reviewer #4: Yes

Reviewer #3: (No Response)

Reviewer #4: After a thorough review of the authors’ revisions, I find that they have adequately addressed the requested changes. The manuscript now meets the necessary standards and is acceptable for publication in its current form.

**Do you want your identity to be public for this peer review?** For information about this choice, including consent withdrawal, please see our Privacy Policy

Reviewer #3: No

Reviewer #4: Yes: Roghieh Nooripour
